# NOXA Is Important for *Verticillium dahliae*’s Penetration Ability and Virulence

**DOI:** 10.3390/jof7100814

**Published:** 2021-09-28

**Authors:** Xiaohan Zhu, Mohammad Sayari, Md. Rashidul Islam, Fouad Daayf

**Affiliations:** 1Department of Plant Science, Faculty of Agricultural and Food Sciences, University of Manitoba, 222 Agriculture Building, Winnipeg, MB R3T 2N2, Canada; huhu772@hotmail.com (X.Z.); Mohammad.Sayari@umanitoba.ca (M.S.); 2Department of Plant Pathology, Faculty of Agriculture, Bangladesh Agricultural University, Mymensingh 2202, Bangladesh; rasha740177@yahoo.com

**Keywords:** verticillium wilt, Nox, ROS

## Abstract

NADPH oxidase (Nox) genes are responsible for Reactive Oxygen Species (ROS) production in living organisms such as plants, animals, and fungi, where ROS exert different functions. ROS are critical for sexual development and cellular differentiation in fungi. In previous publications, two genes encoding thioredoxin and NADH-ubiquinone oxidoreductase involved in maintaining ROS balance were shown to be remarkably induced in a highly versus a weakly aggressive *Verticillium dahliae* isolate. This suggested a role of these genes in the virulence of this pathogen. NoxA (NADPH oxidase A) was identified in the *V. dahliae* genome. We compared in vitro expression of *NoxA* in highly and weakly aggressive isolates of *V. dahliae* after elicitation with extracts from different potato tissues. *NoxA* expression was induced more in the weakly than highly aggressive isolate in response to leaf and stem extracts. After inoculation of potato detached leaves with these two *V. dahliae* isolates, *NoxA* was drastically up-regulated in the highly versus the weakly aggressive isolate. We generated single gene disruption mutants for *NoxA* genes. *noxa* mutants had significantly reduced virulence, indicating important roles in *V. dahliae* pathogenesis on the potato. This is consistent with a significant reduction of cellophane penetration ability of the mutants compared to the wild type. However, the cell wall integrity was not impaired in the *noxa* mutants when compared with the wild type. The resistance of *noxa* mutants to oxidative stress were also similar to the wild type. Complementation of *noxa* mutants with a full length NoxA clones restored penetration and pathogenic ability of the fungus. Our data showed that NoxA is essential for both penetration peg formation and virulence in *V. dahliae*.

## 1. Introduction

Potato early dying (PED) is a common problem in potato (*Solanum tuberosum*) production [1]. The yield loss caused by PED can be up to 30–50% of total production [2,3,4]. The primary causal agent of PED are two different *Verticillium spp.*, *Verticillium dahliae* Kleb and *Verticillium albo-atrum* Reinke & Berthold [5,6]. *V. albo-atrum* was first identified on potato by Reinke and Berthold in 1879 while *V. dahliae* was firstly identified on dahlia (*Asteraceae* family) by Klebahn in 1913 [5]. *V. dahliae* can produce resting structure-microsclerotia that retain viability in the soil for 10–15 years [7,8]. *V. dahliae* interacts with the root-lesion nematode *Pratylenchus penetrans* (Cobb) Filipjev & Schuur. Stekh., which has been shown to facilitate PED in North America [7,9,10]. Root-lesion nematodes cause an increase in root branching and enhance contact between *V. dahliae* and the root facilitating vascular colonization [7,11]. Plants inoculated with both *V. dahliae* and *P. penetrans* showed a higher percentage of root-tip infection than those inoculated with *V. dahliae* alone, indicating that *V. dahliae* and *P. penetrans* may interact to affect host physiology and plant defense responses [11].

One of the key factors in controlling PED is to reduce the primary inoculum by preventing the germination of microsclerotia and decreasing their abundance in soil [12]. Crop rotation has been shown to be an effective control management practice for microbial pathogens in many other economic crops [13,14,15,16,17]. However, *V. dahliae* infects more than 200 dicotyledonous plant hosts [6,18], including many economically value dated crops such as potato, tomato (*Lycopersicon esculentum*), cabbage (*Brassica oleracea*), eggplant (*Solanum melongena*), cauliflower (*Brassica oleracea*), cotton (*Gossypium hirsutum*), bell pepper (*Capsicum annuum*), chili pepper (*Capsicum annuum*), and lettuce (*Lactuca sativa*) [19], which renders crop rotation an ineffective method for controlling this disease.

To date, there is no known treatment that can completely inhibit the disease PED or help recover the yield of affected crops [20]. Green manures such as Austrian winter pea (*Pisum sativum*), broccoli, Sudan grass (*Sorghum vulgare*) and corn (*Zea mays*), could suppress symptoms by 60–70% and partly recover potato yields [21,22]. Chitin and chitosan, which originate from marine crustaceans, also help protect plants from pathogen infections via activation of the host defense [23]. However, the effect of green manure is usually unpredictable and inconsistent, and the mechanisms of suppression of *V. dahliae* microsclerotia is different under various conditions [12]. Moreover, green manures do not decrease and may even raise the amount of *V. dahliae* microsclerotia in the soil [24]. Broccoli (*Brassica oleracea* Italica group) residues suppress *V. dahliae* and reduce both the amount of microsclerotia in soil [25,26] and wilt symptoms in cauliflower [27]. Glucosinolates, phenolic compounds, and lignin in broccoli may be critical for the suppression of *V. dahliae*, which, however, may not exhibit the same effect on other crops [20,28]. 

Many biocontrol agents can also reduce microsclerotia and Verticillium wilt, but yields of infected crops were only partly recovered compared to healthy plants [29,30,31,32]. In tomato, a *Ve*-gene that modulates resistance to race 1 of *V. dahliae* and *V. albo-atrum* was identified and introduced into other tomato cultivars [33,34]. Introduction of the wild relative eggplant (*Solanum torvum*) *StoVe1* gene into potato partially increased resistance to *V. dahliae* [35]. In potato, a *StVe1* locus was identified on chromosome 9, however this is a quantitative trait locus that contains multiple genes (at least 11 genes) and it is still unclear if a single gene or multiple genes provide resistance to *V. dahliae* and *V. albo-atrum* [36,37]. Soil fumigation has been reported to be the most effective strategy for controlling *V. dahliae* [7,38,39,40,41]. The high cost along with environmental and health problems associated with some fumigants have made it necessary to find alternative methods to control the disease [7,42,43,44]. 

Since none of the current control methods represents an ideal choice for controlling Verticillium wilt, it is important to find an alternative strategy to reduce the number of microsclerotia and inhibit their germination and, consequently, reduce penetration and inhibit host colonization by *V. dahliae*. In the past 14 years, research has shown the important functions of reactive oxygen species (ROS) in (1) fungal pathogen penetration and host colonization; (2) normal spore germination; (3) mycelium polarized growth and differentiation; (4) sexual development and fruiting body formation; (5) nutrition transformation under starvation conditions; and (6) germination of the pathogen resting structure, such as sclerotia in *Sclerotinia sclerotiorum* [45,46,47,48,49,50,51,52,53]. ROS can be generated by both non-enzymatic and enzymatic systems [54]. Mitochondria are the primary source of non-enzymatic ROS production [54]. NADPH oxidase (Nox) is the main source of enzymatic ROS production [55]. In various organisms, the Nox protein with FADH_2_ and heme as cofactors can transport the electrons from NADPH to oxygen to produce superoxide [55,56]. Fungi contain one or more of three types of Nox homologues: NoxA, NoxB, NoxC [57]. The structures of NoxA and NoxB are similar to mammalian NADPH oxidase subunit gp^91phox^ [48,58,59], with the exemption of an additional 40 amino acids motif at the N-termini of NoxB [48,59]. The structure of fungal NoxC is similar to mammalian Nox5 [60]. The full function of NoxA and NoxB requires formation of a Nox complex for activation [61]. However, there is no evidence to show that NoxA or NoxB could synchronously interact with all the regulatory subunit in fungi [55]. 

In *Magnaporthe oryzae*, Nox-dependent ROS is essential for full development of the infectious cell, called an appressorium (fungal infectious cell), and pathogenicity on rice [53]. In *Podospora anserina* and *Neurospora crassa*, the *nox1* mutants cannot differentiate the fruiting bodies properly and produce significantly less ascospores than the wild type [48], while *nox2* mutants can produce ascospores but none of them can germinate [48,51]. In *Sclerotinia sclerotiorum*, SsNox1 and SsNox2 have been identified and are responsible for ROS production [49]. Both SsNOX1 and SsNOX2 are required for sclerotial formation, while SsNOX1 is also essential for virulence [49]. In *Fusarium graminearum*, NoxA is critical for ROS production during perithecia development and ascospore production and pathogenic development on wheat [50]. In *Claviceps purpurea*, Nox1 is essential for conidial germination, mature sclerotia development and virulence [45]. In *Botrytis cinerea*, functional characterization of *BcNoxA* and *BcNoxB* showed that ROS generated by both NoxA and NoxB is essential for virulence and development of sclerotia [62]. In *Aspergillius nidulans*, deletion of *NoxA* affected the sexual development by blocking the formation of mature cleitothecia fruiting bodies [58]. In *Epichloë festucae,* NoxA as well as its signal regulator RacA, and NoxR play critical roles in controlling mutualistic symbiotic interaction between *E. festucae* and the host perennial ryegrass [47]. Recent studies showed VdNoxB, identified in a *V. dahliae* cotton isolate, was required for Ca^2+^ accumulation in hyphopodia via NoxB-produced ROS and activity regulation of the transcription factor VdCrz1 in the control of the penetration peg development on cotton [63].

Taken together, this indicates that Nox enzyme-producing ROS in various fungal species play important roles in penetration, colonization, and pathogenic development in the host, as well as development of resting or over-wintering structures [45,46,47,48,49,50,51,52,53]. The management of PED in potato relies on the control of the primary inoculum and reduction of microsclerotia in the soil. Therefore, we speculate that ROS generated by the Nox family are important for the interaction with potato root penetration. El-Bebany and Rampitsch [64] identified two proteins, Thioredoxin and NADH-ubiquinone oxidoreductase, which function in the maintenance of ROS balance in the cell [64,65,66,67]. In a proteomic analysis, both proteins were only detected in a highly aggressive isolate of *V. dahliae* but not in a weakly aggressive one [64]. According to our unpublished data, NADPH oxidase (NOX) was also involved in the pathogenicity-related pathway in *V. dahliae*. All of these indicate that ROS in *V. dahliae* may be critical for pathogenicity-related processes. NoxB has been proven to be involved in penetration peg formation in a cotton isolate [63]. In the present study, we aimed to investigate the function of the *NoxA* gene in the highly aggressive *V. dahliae* isolate Vd1396-9 during potato infection. According to Klosterman and Subbarao [68], *V. dahliae* contains three *Nox* isoforms: *NoxA* (VDAG_06812.1), *NoxB* (VDAG_09930.1) and *NoxC* (VDAG_00032.1) [68]. The objectives for this study were to: (1) investigate the transcriptional activity of *V. dahliae*’s *NoxA* gene during both elicitation with host plant tissue extracts and during infection; (2) generate *NoxA* gene mutants in *V. dahliae* and analyze their phenotypes; (3) assess the roles of the *NoxA* gene in pathogen virulence and during the interaction with potato; and (4) determine its roles in cell wall biosynthesis and response to oxidative and osmotic stress.

## 2. Materials and Methods

### 2.1. V. dahliae Isolates and Plant Materials

Vd1396-9 and Vs06-07 have been identified as highly and weakly aggressive *V. dahliae* isolates, respectively [69,70]. They were isolated from a potato tuber and sunflower stem tissue, respectively [69,70]. Both isolates were grown on potato dextrose agar (PDA) media at 23 ± 1 °C for 21 days. Culture plates were flooded with sterilized water, and then spores were collected for each isolate and counted using a hemacytometer counting chamber (Fisher Scientific, Hampton, NH, USA), followed by concentration adjustment according to different experimental requirements.

The potato cultivar Kennebec, which is susceptible to Verticillium wilt, was used in this study [69]. Plants were grown in a mix of sand, soil and peat moss (12:4:1) in a greenhouse growth room at a 22/18 °C day/night temperature regimen with a 16/8 h light/dark photoperiod.

### 2.2. NoxA Genes Expression in Response to Infection and During Elicitation with Potato Tissue Extracts 

*NoxA* expression in both Vd1396-9 and Vs06-07 were measured during infection on detached Kennebec potato leaves following the method described by Zhu and Soliman [71]. Briefly, conidia of Vd1396-9 and Vs06-07 were washed from PDA plates and then adjusted to the concentration of 3 × 10^7^ conidia/mL, after which they were inoculated on detached susceptible potato leaves (Kennebec). Samples of detached leaves were then taken at one, three, five and eight days after inoculation (DAI). Additionally, gene expression in both isolates under elicitation of Kennebec potato leaves, stems, or roots extracts was determined by qRT-PCR following the method described by Zhu and Soliman [71] and El-Bebany, andHenriquez [72]. Briefly, 10^8^ conidia were cultured in Czapek-Dox Broth (CDB) media (Difco Laboratories, Sparks, MD, USA) for one week. Each isolate was then treated with the addition of 1 mL of potato leaf, stem, or root extract for one week. Samples of fungal mycelium were collected and subjected to RNA extraction and Real time PCR analysis following the recommended protocols of the Omega Fungal RNA kit (Omega Bio-Tek, Inc., Norcross, GA, USA) and the SsoFast EvaGreen Super mix, respectively (Bio-Rad Lab, Philadelphia, PA, USA). 

### 2.3. Gene Disruption and Complementation of V. dahliae

For the knockout study, *NoxA* (VDAG_06812.1) a gene T-DNA insertion construct was created based on the pDHT vector [73] following the description specified by Zhu and Soliman [71], with primers listed in (Table 1). To be more specific, the open reading frame (ORF) of *NoxA* was amplified from genomic DNA of strain Vd1396-9 with specific primers flanked with restriction sites of HindIII (Table 1). The DNA fragment was then cloned into the binary vector pDHT and mutagenized using the EZ::TN transposon system (Epicentre Technologies, Madison, WI, USA). The constructs were transformed into Vd1396-9 conidia mediated by *Agrobacterium tumefaciens* following the description by Zhu and Soliman [71] and Dobinson and Grant [74]. Transformants were selected according to the method described by Zhu and Soliman [71] in PDA media containing hygromycin B. The positive gene insertion mutants were confirmed by PCR with primer pairs NoxA-UA-F and NoxA -HindIII-R (Table 1).

The single locus insertion in the *V. dahliae* genome of the *noxa* mutants, and gene duplication of the NoxA in the genome, were both confirmed by southern blot with specific probe (amplified by primers NoxA-HindIII-F and Hph-YG-F, Table 1) and restriction enzyme (XhoI). For DNA extraction, the *V. dahliae* mycelia were collected after a one-week culture in CDB liquid media. The DNA was extracted according to the protocol described by Al-Samarrai and Schmid [77]. The southern blot, probe hybridization, detection and signal visualization were processed following the description by Zhu and Soliman [71] and Maruthachalam et al. (2011).

The *NoxA* complementation strain was constructed by using the primers NOXA-F-1-A and NOXA-R-4-A to amplify the *NoxA* gene from the genomic DNA of *V. dahliae*. The amplicon was then ligated into the Geneticin containing vector PC-g418-YR (Addgene ID—61767). PCR primer sequences used are shown in Appendix A. PCR conditions were as follows: 95°C 1 min, followed by 35 cycles of 95 °C for 30 s; 60 °C for 60 s; 65 °C for 50 s, followed by final extension at 65 °C for 15 min.

### 2.4. Growth Rate and Conidiation of Noxa Mutants

The *V. dahliae* mutants for the *NoxA* gene (*noxa-im-1*, *noxa-im-5*, and *noxa-im-7*), the ectopic insertion strains for *NoxA* gene (*NoxA-Ect-3*) (randomly inserting in *V. dahliae* genome but without replacing the original *NoxA* ORF), and the empty vector control insertion strain (EVC; an empty pDHt vector, instead of mutation vector, was transformed into Vd1396-9) together with the wild type Vd1396-9, were grown on PDA for 14 days. The growth rate of the colony and the conidia concentration were determined according to the description of Zhu and Soliman [71].

### 2.5. The Pathogenicity Analysis of Noxa Mutants and Noxa Complementation Strain

Potato plants (cv. Kennebec) were grown in soil-less mix (LA4—SunGro Horticulture, Agawam, MA, USA) for one week, and then plants were gently uprooted and approximately one cm long root tips were trimmed and immediately placed in conidial suspensions at a concentration of 10^6^ conidia/mL using the following mutants: *noxa-im-1*, *noxa-im-5*, *noxa-im-7*, *NoxA-Ect-3*, EVC, disruptants complemented with *NoxA* genes noxa_im_1_comp, as well as the wild type. Sterile water was used as a control treatment. After a 30-s inoculation treatment with a conidial suspension, infected plants were re-planted in a pasteurized sand, soil and peat moss mixture with a ratio of 16:4:1. Each treatment contained five biological replicates. The total area under a disease progress curve (AUDPC) of percentage infection and disease severity, together with plant growth rate were determined according to Zhu and Soliman [71]. AUDPC is a helpful measurable synopsis of disease intensity across time, for comparison over years, sites, or controlling methods. In contrast, disease severity is defined as the area of diseased plant tissues compared to the whole plant. The stem vascular discoloration was recorded in the last week of assessment according to Alkher and El Hadrami [69].

### 2.6. Cell Wall Biosynthesis and Response to Stress Conditions

Calcofluor white can be used as a specific fungal chitin marker, which combines with fungal polysaccharides and changes the assembly of chitin fibrils in the fungal cell wall [78]. To assess the role of the *NoxA* gene in cell wall biosynthesis, *noxa,* mutants and wild type *V. dahliae* were cultured on solid CDB medium containing calcofluor white with concentrations at 0, 50, 90, or 150 µg.mL^−1^. Oxidative stress resistance was determined on the mutants and wild type by culturing strains on solid CDB medium with H_2_O_2_ concentrations at 10 mM, 20 mM, and 30 mM. To determine the response to osmotic stress, mutants and wild type were cultured on solid CDB medium containing 0.8 M NaCl. The strain diameter on various treatments were measured after culturing for 10 days to estimate the inhibition rate under various stress levels following the description of Guo and Chen [79].

### 2.7. Penetration and Germination Abilities of Noxa Mutants on Cellophane Membrane

The fungal penetration assay was conducted using a cellophane membrane following the method described by Wang, Mogg [50]. Conidia suspension of mutants and wild type *V. dahliae* strains (10^5^ conidia/mL) were cultured on cellophane membranes placed on solid CDB media. After culturing for either 5 or 21 days, the cellophane membranes were removed, and the cultured plates were placed at 23 ± 1 °C for an additional 4 days. Isolates that successfully penetrated mycelium exhibited growth on the solid CDB medium.

The germination ability of *noxa* mutants, the disruptant complemented with *NoxA* genes and the wild type strain were observed under microscopy at 24 h, after a conidia suspension (10^5^ conidia/mL) of each isolate was placed on cellophane membranes laying on top of solid CDB medium. 

The ability of all isolates to form penetration pegs were also observed under microscopy, 72 h after a conidial suspension was placed on cellophane membranes on top of a solid CDB medium. 

### 2.8. Formation of Conidiophores of Noxa Mutants

*noxa* mutants and the wild type *V. dahliae* strain were cultured on PDA media for 2 weeks, after which a hole (1 cm) was cut and observed following 48 h culturing under the same conditions. The conidiophores of each isolate were observed on the edge of the hole under microscopy.

### 2.9. Statistical Analysis 

SAS Statistical Analysis Software (SAS Institute, Cary, NC, USA; release 9.1 for Windows) with the PROC MIXED program was used for statistical analysis of all data in this study. All data qualified for normal distribution with Shapiro–Wilk test (>0.9) determined by the PROC UNIVARIATE program. They also qualified for homogeneity established on comparison residuals and studentized residual critical values [80]. Some series of data were treated with Log^10^ transformation before analysis when necessary. Mean values were separated according to least squared means and results were assigned a group of letters using the macro PDMIX800.sas [81] with α = 0.05. Results assigned with different letters indicate significant differences between tests (*p* < 0.05).

## 3. Results

### 3.1. Expression of NoxA Gene in V. dahliae in Response to Potato Extracts and Infection

ROS plays an important role in the virulence development of several phytopathogens [50,53]. The role of ROS produced by *NoxA* gene was investigated through their expression in both *V. dahliae* highly and weakly aggressive isolates Vd1396-9 and Vs06-07 during the infection in potato or under elicitation with potato tissue extracts. The expression of *NoxA* was higher in the weakly aggressive isolate Vs06-07 under the elicitation of leaf and stem extracts (Figure 1A). During the infection on detached potato leaves, *NoxA* was significantly induced in the highly aggressive isolate compared to that in the weakly aggressive one (Figure 1B). 

### 3.2. Generation of Gene Insertion Mutants for NoxA Family Member and Transformant Complemented with Full Length NoxA Genes

To investigate the functions of the *NoxA* gene in *V. dahliae* during its interaction with potato, an individual gene insertion mutant was generated. Sequencing of the generated mutants indicated that insertion events occurred at No. 789 bp of *NoxA* ORF in the gene disruption mutants. Transformants were firstly screened by PCR, which identified 13 positive transformants for *noxa* gene mutants (Figure 2A,B). To determine the insertion number of the DNA cassette containing the hygromycin resistant gene (*hph*) and gene duplication of *NoxA* in the *V. dahliae* genome, the positive transformants were randomly selected for Southern blot. The number of the DNA cassette insertion and gene duplication of *NoxA* was determined using the same probe containing part of the DNA fragment of *NoxA* ORF and part of the DNA fragment of the hygromycin resistant gene. Our results showed that all seven of the selected transformants of *noxa* mutants (Figure 2C) were single-insertion mutants for the corresponding gene, and *NoxA* gene were in single copy in the *V. dahliae* genome (Figure 2C). All mutants confirmed by PCR and Southern blot analysis made up the population from which individual mutants were randomly selected for pathogenicity tests.

After complementation, we could amplify the full length of *NoxA* in the representative complementation strains (Appendix A). A total of 25 *NoxA-im-1-comp* exhibited wild type characteristics such as the ability to produce penetration peg on cellophane membrane as well as to infect potato plants, (Appendix A). Representatives of complemented strains were confirmed by ability to grow on selective media as well as sequencing.

### 3.3. Growth Rate and Conidiation of Noxa Mutants

The growth rate and spore production of the *noxa* mutants were assessed on PDA medium. Pathogenicity was tested on the susceptible potato cultivar Kennebec. The growth rate, colony morphology, spore production, and microsclerotia formation of *noxa* mutants (*noxa-im-1*, *noxa-im-5*, and *noxa-im-7*) were not significantly different from the ectopic control *NoxA-Ect-3*, empty vector control (EVC), and wild type Vd1396-9 (Figure 3).

### 3.4. The Pathogenicity Analysis of Noxa Mutants and Noxa Complementation Strain

The total AUDPC of infection and disease severity, as well as the vascular discoloration rate of infected potato stems caused by *noxa* mutants (*noxa-im-1*, *noxa-im-5*, and *noxa-im-7*) were dramatically reduced on three sets of experiments conducted from 2016 to 2018 (Figure 4, Figure 5 and Figure 6). The growth rate of the potato plants inoculated with the mutants was similar to that of the water control treatment, but significantly higher than that of those inoculated with the ectopic control, wild type Vd1396-9, and EVC (Figure 4C, Figure 5C and Figure 6C). These results indicate that disruption of the *NoxA* can significantly reduce the virulence of *V. dahliae* on the potato cultivar. The pathogenicity of the fungus was restored in the complemented strains *NoxA-im-1-comp* (Appendix A).

### 3.5. Cell Wall Biosynthesis

There was no significant difference between *noxa* mutants and the wild type in response to Calcofluor white treatment (Appendix A). This indicates that cell wall biosynthesis was not affected in *noxa* mutants.

### 3.6. Resistance to Oxidative Stress and Osmotic Stress

To determine the response to oxidative stress and osmotic stress, *noxa* mutants and the wild type strains were cultured on solid CDB medium with varying H_2_O_2_ concentrations as well as with 0.8 M NaCl. There was no significant difference between *noxa* mutants and the wild type with respect to oxidative stress and osmotic stress (Appendix A).

### 3.7. The Penetration, Germination Ability as Well as Conidiophore Formation of the Noxa Mutants and Complemented Mutants 

To further assess the effect of *noxa* mutants on virulence, the penetration ability of mutants and wild typeisolates was determined on a cellophane membrane. Following both 5 and 21 days-post-inoculation, *noxa* could not penetrate the cellophane membrane as the wild type did (Figure 7). This indicates that the penetration ability was affected in *noxa* mutants compared to the wild type. The penetration ability of *noxa* complemented strains were restored and they could penetrate through the cellophane membrane after five days (Appendix A). This observation confirmed the fact that NoxA is responsible for the penetration ability of *V. dahliae*. Furthermore, the germination ability of *noxa* mutants and the wild type strain were observed 24 h-post-inoculation (HPI) and all tested isolates can normally germinate on cellophane membrane (Figure 8A). However, at 72 HPI, all three *noxa* mutants failed to form the penetration peg in the hyphopodium cell, while, the control strain including NoxA-Ect-3, EVC and the wild type strain Vd1396-9, formed penetration peg in the hyphopodium cell (Figure 8B). After complementation, *NoxA-im-1-comp* could produce penetration peg at 72 HPI (Appendix A). Conidiophore morphology was similar between *noxa* and the wild type strain (Figure 8C).

## 4. Discussion

*V. dahliae* causes wilt symptoms in more than 200 dicotyledonous plant species, and in potato it contributes to potato early dying (PED) [6,7]. The management of this disease depends on crop rotation, green manure, and soil fumigation; however, these methods are either costly or ineffective [12,19,27,29,38]. Tomato plants with *Ve*1-gene showed resistance to race 1, not race 2, of *V. dahliae* and *V. albo-atrum* [33,34]. A quantitative trait locus (QTL) containing at least 11 different homologues (leucine- rich repeat (LLR) protein) of *StVe1* was identified in chromosome 9 in tetraploid potato, but whether single or multiple copies of these homologues provide resistance to *V. dahliae* and *V. albo-atrum* is still not clear [36,37]. In past years, the disease resistance cultivars of potato were not applied in a wild range [82]. Resistance is defined as restricting the development of pathogen or disease symptoms on the host [82]. In other words, disease resistance does not entail killing the pathogen completely, but is an active process that could restrict the pathogen in the host [82]. One of the most effective ways to manage this soil-borne disease may be by controlling the initial inoculum, both in the soil and during pathogen infection on the host [82]. 

El-Bebany and Rampitsch [64] conducted comparative analyses on two *V. dahliae* isolates differing in virulence, and detected two proteins, Thioredoxin and NADH-ubiquinone oxidoreductase, only in the highly aggressive isolate, while they were lacking in the weakly aggressive one, with the former protein playing a role in ROS cleavage and the latter functioning in non-enzymatic ROS production in mitochondria [64,65,66,67]. Knowing that ROS generated by Nox proteins were important for sexual development and host penetration [50,56,63,83,84]. Furthermore, NoxB also proved to be important in *V. dahliae* and responsible for the normal formation of the penetration peg and virulence [63], so we investigated the functions of *NoxA* genes in *V. dahliae*. 

In response to different potato tissue extracts, *NoxA* genes transcriptionally expressed in weakly aggressive isolate under stem and leaf extracts. However, it expressed more in the highly aggressive isolate during infection of detached leaves. This may indicate that the function of the *NoxA* gene in different processes of infection or interaction with the host plant may be various. The increasing transcriptional activity of *NoxA* in the highly aggressive *V. dahliae* is in line with the findings in *C. purpurea* that expression of *CpNOX1* increases during infection in planta and reaches a maximum at a later infection stage [45]. In other fungi, the *Nox* family genes usually play roles in different processes of cell differentiation [57]. Although these genes have similar structure and function, they still have different properties in various cellular processes such as penetration, production and germination of ascospores, and pathogenicity [48,53,84]. 

The expression pattern of the *NoxA* in highly versus weakly aggressive isolates was a surprise since this gene is not so much induced upon infection. However, looking at the genome of *V. dahliae* showed that there are other Nox genes in the genome of this fungus (data not shown). So, one hypothesis would be that other Nox genes might compensate for NoxA, as the noxA mutant has no defects in the cell wall nor an alteration in sensitivity to H_2_O_2_. However, since mutants of *NoxA* had much lower virulence than their wild type counterpart, it seems that in *V. dahlia, NoxA* must play critical roles during specific infection processes. This defect in virulence is similar to *nox1* and *nox2* mutants in *M. oryzae*, *noxa* and *noxb* mutants in *F. graminearum* and *B. cinerea*, as well as *nox1* mutant in *C. purpurea* [45,50,53,62]. This apparently indicates that NoxA would play important roles in the pathogenicity of *V. dahliae*. The homologue of Nox1 or NoxA in *P. anserina*, *N. crassa*, *F. graminearum* and *A. nidulans*, would regulate sexual development [48,50,51,58]; however, since *V. dahliae* has no known sexual stage [85], we were unable to study the roles of *NoxA* in this process. The growth rate, spore production, and formation of microsclerotia were not affected in the mutants of *NoxA*. This is in contrast to the observation on other ascomycetous fungi such as *M. grisea*, *P. anserina*, *N. crassa*, *S. sclerotiorum*, *F. graminearum*, *C. purpurea*, and *B. cinerea* where homologues of mammalian gp^91phox^ regulate spore production and germination, and resting structure development [45,48,49,50,51,53,62]. *NoxA* in *V. dahliae* was not involved in cell wall biosynthesis, which is in contrast to *Nox1* in *M. oryzae* [53]. *NoxA* in *V. dahliae* are important for penetration on cellophane membrane, as *noxa* mutants could not breach the cellophane membrane after five- and 21-day-inoculation. Furthermore, the reason that *noxa* lost penetration ability is that mutants can no longer form the normal penetration peg on the cellophane membrane. In *M. oryzae*, *P. anserina*, *F. graminearum*, and *B. cinerea*, homologues of Nox also regulate penetration on the host [48,50,53,62,84]. In another study involving a *V. dahliae* cotton isolate, VdNoxB and tetraspanin VdPls1 seemed to function in a co-located manner in hyphopodia for penetration peg formation [63]. VdPls1 was proven to be the adaptor protein for VdNoxB and the one to control its activity, while both regulated the accumulation of intracellular Ca^2+^ in hyphopodia tip [63]. This was also proven to be important for penetration peg formation [63]. However, the relationship and cooperation between NoxA and NoxB is unknown. In addition to the loss of the ability to form the normal penetration peg, another explanation may be that homologues of Nox may regulate cellulose degradation to control the penetration process in the host. This has been proven by studies on Nox1 and Nox2 in facilitating cellulose degradation in different manners and affecting the penetration ability as well in *P. anserina* [84].

## 5. Conclusions

In this study, we investigated the functions of NoxA gene in *V. dahliae* and showed that this gene is one of the possible genes manipulating fungal penetration into the host and in facilitating the virulence during the interaction between *V. dahliae* and potato. Since the other Nox genes can also play roles in the penetration ability and consequently the pathogenicity of the fungus, further studies will focus on decoding the detailed molecular mechanisms for regulating the penetration. This research may help provide more strategies to prevent the initial infection of *V. dahliae* as part of integrated disease management practices to control verticillium wilts.

## Figures and Tables

**Figure 1 jof-07-00814-f001:**
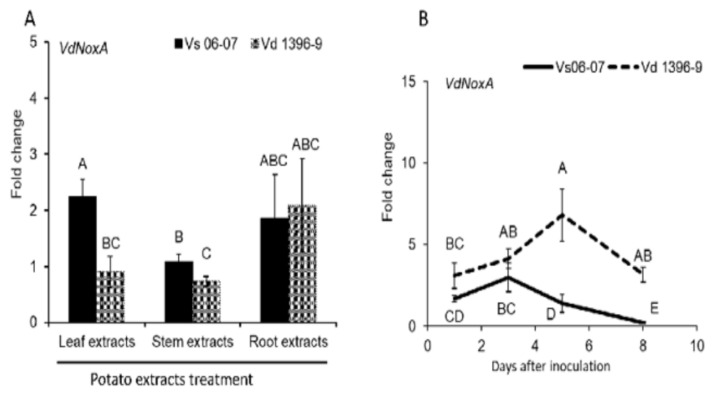
Expression of *NoxA* gene in *V. dahliae* under elicitation or during the infection. (**A**) Expression of *NoxA*, in response to potato leaves, stems, and roots extracts. The highly (Vd1396-9) and weakly (Vs06-07) aggressive isolate were cultured in liquid CDB medium added with potato leaves, stems or roots extracts. Sterilized distilled water was added into culture medium as a control treatment. Both Vd1396-9 and Vs06-07 cultured in the CDB medium with water were used as calibrators. The *V. dahliae* Histone H3 gene was employed as the internal control for normalizing all qRT-PCR data. The expression data for each gene in selected isolate in response to treatments were analyzed with the 2^−ΔΔC^_T_ method, in relation to water treatment as the control group. The bars shown as mean values (*n* = 3) with different letters were significantly different between treatments (*p* < 0.05). Error bars refer to standard error. (**B**) Expression of NoxA during the infection on detached Kennebec potato leaves. Four to six pieces of four-week-old Kennebec potato detached leaves from different individual plant were combined as one sample after inoculation by highly (Vd1396-9) or weakly (Vs06-07) aggressive isolate using 10^8^ conidia/mL. Sterilized distilled water was mocked to inoculate the detached leaves as the control treatment. Three combining samples were prepared for each treatment at each time point (one, three, five and eight days after inoculation, DAI). Both Vd1396-9 and Vs06-07 cultured in CDB medium were used as the calibrators. The *V. dahliae* Histone H3 gene was employed as the internal control for normalizing all qRT-PCR data. The expression data for each gene during the infection were analyzed with the 2^−ΔΔC^_T_ method, in relation to isolates cultured in CDB medium as the control group. The point values showed as mean values (*n* = 3) with different letters were significantly different between treatments (*p* < 0.05). Error bars refer to standard error. The LSD post hoc test was performed to determine which differences are significant among the different treatments. The bars/points with different letters are significantly different.

**Figure 2 jof-07-00814-f002:**
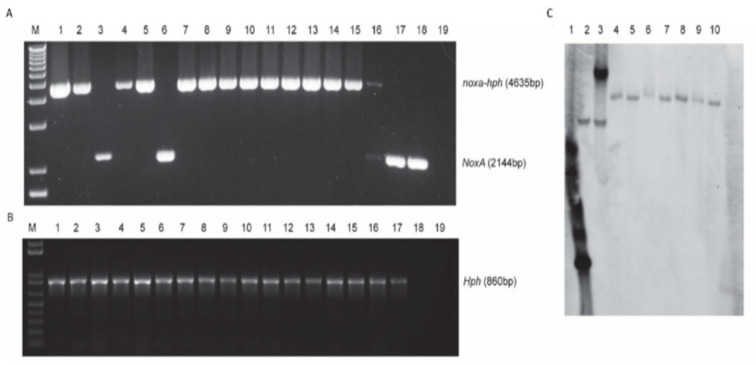
Identification of *noxa* mutants by PCR and southern blot. (**A**,**B**) PCR analysis of transformants for *NoxA* gene insertion. Lane M represents the DNA markers (1 Kb Plus DNA Ladder, Invitrogen, Waltham, MA, USA), lane 1 to 17 represent the transformants, lane 18 represents the genomic DNA of wild type strain Vd1396-9, and lane 19 represents the negative water control for PCR. *noxa-hph*: *NoxA* gene disrupted by inserting a DNA cassette containing both a chloramphenicol resistance gene and a hygromycin phosphotransferase gene in the original *NoxA* ORF region; *Hph*: Hygromycin phosphotransferase gene. (**C**) Southern blot analysis of positive transformants of *noxa* mutants. Lane 1 represents the probe, Lane 2 represents wild type strain Vd1396-9, lane 3 represents ectopic control of *NoxA* insertion, and lane 4 to 10 represent positive transformants of *noxa* mutants. NoxA southern blot probe amplified by primers NoxA-HindIII-F and Hph-YG-F (Table 1), which contain part of DNA fragment of NoxA ORF and part of DNA fragment of the hygromycin resistant gene.

**Figure 3 jof-07-00814-f003:**
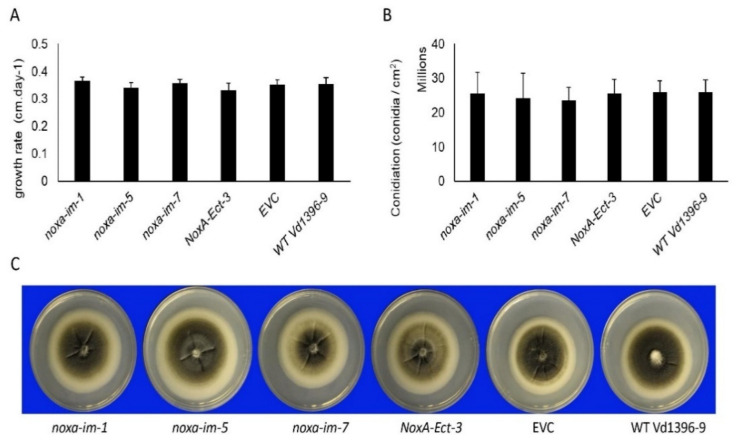
Phenotypic analysis of *noxa* mutants on PDA medium. (**A**) The growth rate of *noxa* mutants. (**B**) The conidiation of *noxa* mutants. (**C**) The colony phenotype of *noxa* mutants. The bars shown as mean values (*n* = 8) for growth rate experiment and (*n* = 5) for conidiation experiment. No differences were observed between different isolates (*p* < 0.05). Error bars refer to standard error.

**Figure 4 jof-07-00814-f004:**
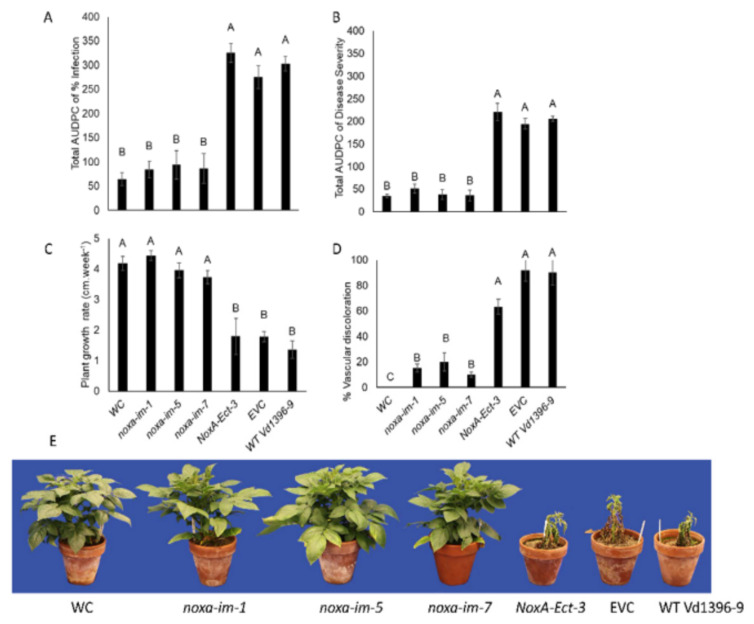
Pathogenicity test of *noxa* mutants on susceptible potato cultivar (Kennebec) in 2016. (**A**) Total AUDPC of percentage of infection. (**B**) Total AUDPC of disease severity. (**C**) Growth rate of potatoes. (**D**) Percentage of vascular discoloration. (**E**) Kennebec potatoes infected by *noxa* mutants at six weeks after infection. Error bars refer to standard error. The LSD post hoc test was performed to determine which differences are significant among the means. Bars (*n* = 4) with different letters indicate significant differences (*p* < 0.05).

**Figure 5 jof-07-00814-f005:**
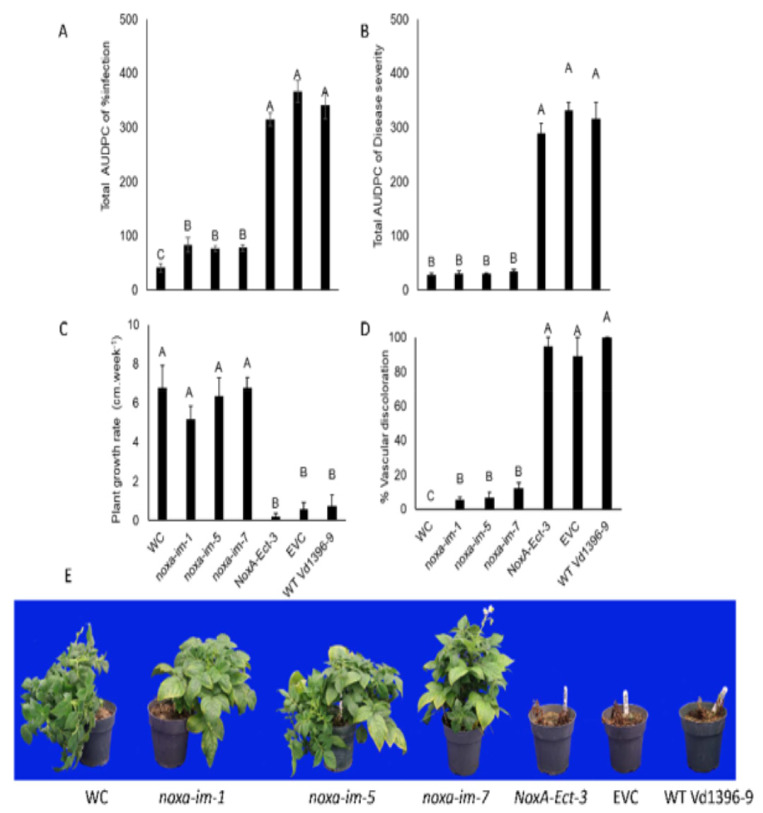
Pathogenicity test of *noxa* mutants on susceptible potato cultivar (Kennebec) in 2017. (**A**) Total AUDPC of percentage of infection. (**B**) Total AUDPC of disease severity. (**C**) Growth rate of potatoes. (**D**) Percentage of vascular discoloration. (**E**) Kennebec potatoes infected by *noxa* mutants at six weeks after infection. Error bars refer to standard error. The LSD post hoc test was performed to determine which differences are significant among variable means. Bars (*n* = 5) with different letters indicate significant differences (*p* < 0.05).

**Figure 6 jof-07-00814-f006:**
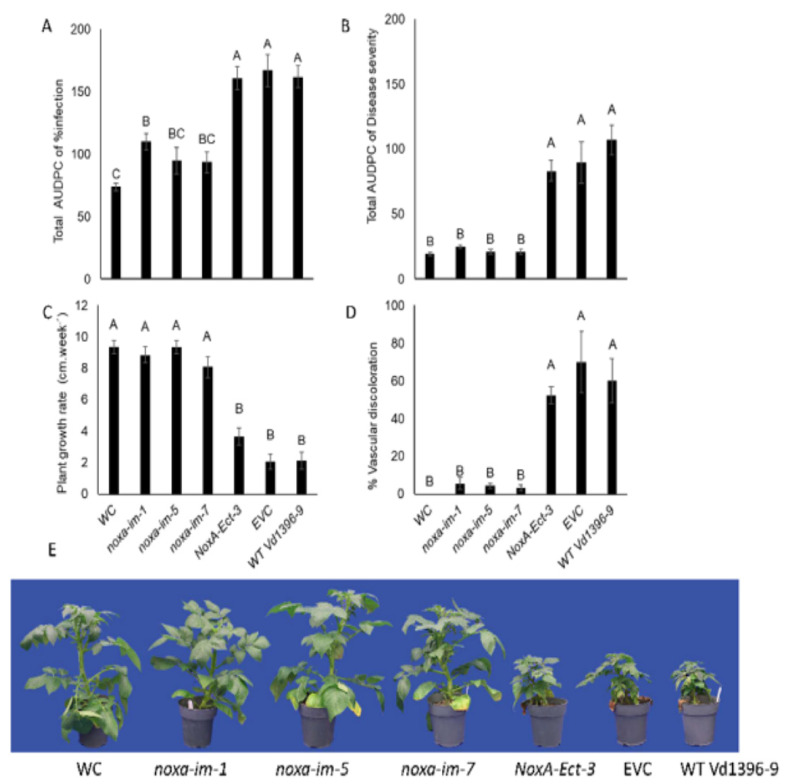
Pathogenicity test of *noxa* mutants on susceptible potato cultivar (Kennebec) in 2018. (**A**) Total AUDPC of percentage of infection. (**B**) Total AUDPC of disease severity. (**C**) Growth rate of potatoes. (**D**) Percentage of vascular discoloration. (**E**) Kennebec potatoes infected by *noxa* mutants at 5 weeks after infection. Error bars refer to standard error. The LSD post hoc test was performed to determine which differences are significant among variable means. Bars (*n* = 6) with different letters indicate significant differences (*p* < 0.05).

**Figure 7 jof-07-00814-f007:**
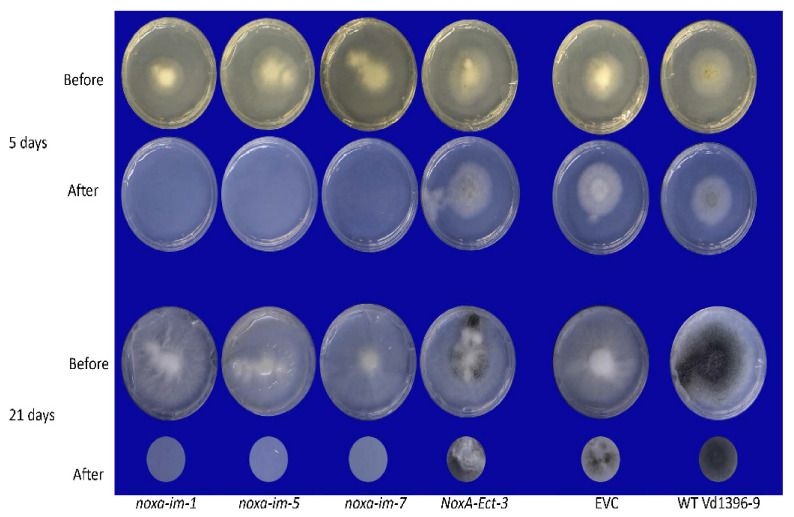
The penetration ability test of *noxa* mutants on cellophane membrane. Each isolate was repeated six times. All isolates were firstly inoculated on solid CDB media covered with a cellophane membrane for five and 21 days (indicated as Before) at 23 ± 1°C, following which the cellophane membranes were removed from the media and maintained under the same conditions for an additional four days (indicated as After). In almost all cases, both wild type and mutant strains could reach the edge of the plates after 21 days of incubation. This does not confirm the penetration ability, but the middle of the plates would only allow those isolates with full penetration ability to grow, as only mycelia with penetration ability can pass through the cellophane sheet from the plate’s center. Therefore, to reduce the confusion, here we only show the middle of the plates after 21 days of incubation.

**Figure 8 jof-07-00814-f008:**
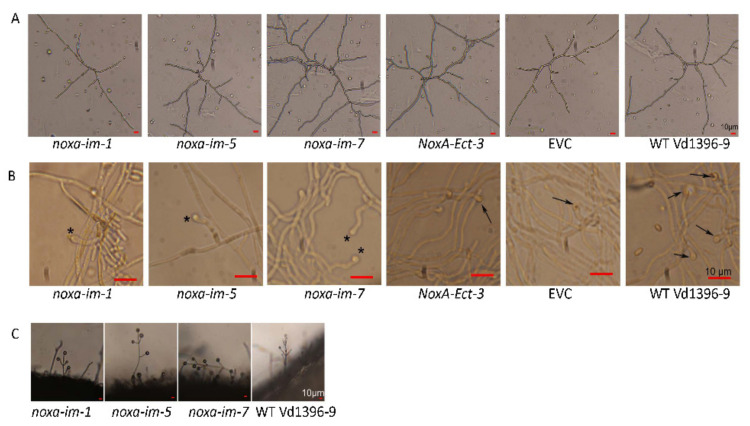
Morphology of *noxa* mutants in germination, penetration, and formation of conidiophores. (**A**) Conidia germination ability test of *noxa* mutants on cellophane membrane at 24 h. The conidia of *noxa* mutants and the wild type strain (WT Vd1396-9) were cultured on cellophane membrane placed on solid CDB media for 24 h, then observed under microscopy. (**B**) The formation of penetration peg test of *noxa* mutants on cellophane membrane at 72 h. The conidia of *noxa* mutants and the wild type strain were cultured on cellophane membrane placed on solid CDB media for 72 h, then observed under microscopy. The penetration pegs are shown as a dark spot in the hyphopodium cell. *NoxA-Ect-3*, EVC, and the wild type strain, can both form the penetration peg, which are indicated by arrows. *noxa* mutants (*noxa-im-1*, *noxa-im-5*, and *noxa-im-7*) formed the hyphopodium cell without penetration peg, which are indicated by asterisks. (**C**) The conidiophore formation test of *noxa* mutants at 48 h. All the *noxa* mutants and the wild type strain were cultured on PDA plates, then a hole was punched on each culture. After 48 h conidiophores were observed on the edge of the hole under microscopy. All bars are equal to 10 µm.

**Table 1 jof-07-00814-t001:** Primers used in generating mutants for Nox family genes.

Primer’s Name	Primer Sequence	Tm (°C)	Accession Number
NoxA-HindIII-F	CCCAAGCTTATGCCTCTCGCCAACCTTT	59.5	VDAG_06812
NoxA-IHindIII-R	CCCAAGCTTTCAGAAATGCTCCTTCCAGAAA	59.3	VDAG_06812
NoxA-UA-F	CCCTCGCCTGACGGGATT	62.7	VDAG_06812
Hph-YG-F [75]	GATGTAGGAGGGCGTGGATATGTCCT	61.5	Hph gene
Hph-F [76]	TCAGCTTCGATGTAGGAGGG	55.6	Hph gene
Hph-R [76]	TTCTACACAGCCATCGGTCC	56.5	Hph gene

Note: Hph gene: hygromycin resistant gene.

## Data Availability

The data that support the findings of this study are available on request from the corresponding author.

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
