# Peer review of "NOXA Is Important for Verticillium dahliae’s Penetration Ability and Virulence"

_jof, 2021, doi:10.3390/jof7100814_

Round 1

Reviewer 1 Report

The manuscript by Zhu et al., describes the importance of NADPH oxidases from Verticillium dahlia in virulence. The fungus genome contains 3 nox genes whose expression varies between highly and weakly aggressive isolates upon elicitation with extracts from different potato tissues. From the 3 nox genes, noxA deletion results in reduced virulence being essential for infection. However, from the previous version of the paper, noxB deletion also results in reduced virulence. In my opinion authors still need to discuss the manuscript findings considering what is known from previous studies. For instance, it  is known that “noxB -mediated ROS production leads to Ca2+ elevation in hyphopodia, infectious structures of V. dahliae, to regulate penetration peg formation during the initial colonization of roots (Zhao Y-L, Zhou T-T, Guo H-S (2016) PLoS Pathog 12(7): e1005793).

When authors claim that “NoxA is responsible for penetration ability of V. dahlia” it is not totally correct since noxB is also responsible for the penetration ability of V. dahlia. Authors need to rephrase this conclusion and discuss it based on what is known.

Figure 2, which probe was used in the southern blot? Please add this information to the legend of the figure.

Figure 3, in my opinion the A in each bar is not needed. Moreso, the figure legend is misleading since the letter is always the same.

Figure 7 and 8 may be moved to supplementary information since there is no phenotype for noxa mutants.

The sensitivity of mutant noxa-im-5 to H2O2 is quite different from that of the other noxa mutants. Please comment this observation.

Letters A, B, C, D and E in figures are not easy to follow. Please add a sentence, in figure legends, explaining how to read the letters code in each figure.

The reviewer mentioned that in Figure 9 and 10, the complemented transformants are not depicted.

We have added results for complementary strains in Supplementary figures 1-4.

I had no access to the supplementary material in this submission.

The reviewer asked why do the plates of 21 days after look so small comparing to the other conditions? Could this be improved?

After growing in plates for 21 days, most of mycelium of both the mutants and wild type strain could grow over the edges of cellophane membrane without penetration, so the whole plates edges would grow a cycle of mycelium. This does not confirm the penetration ability, but middle of the plates would only allow those isolates with full penetration ability to grow as only mycelia with penetration ability can pass through the cellophane sheet from centre of the plates. To reduce the confusion, we only show the middle of the plates here for 21 days.

I understand the explanation provided here and think that this experimental limitation should be referred in the Figure legend.

After removing information about the other nox genes, authors missed some subject-verb agreement. Please carefully check English throughout the manuscript.

Author Response

Response to Reviewers’ Comments

Please note that all changes requested by the reviewer are now shown in highlight throughout the manuscript.

  • The reviewer asked us to discuss the manuscript findings considering what is known from previous studies regarding the role of Nox genes.

Thanks for your comment. Please note that we have added some more discussion to the manuscript as shown in Lines 526-531.

  • The reviewer asked to rephrase the conclusion and discuss it based on what is known.

We have rephrased the conclusion part to make sure that we conclude our results based on what is known as shown in highlight in Lines 539-542.

  • The reviewer asked to add information about the probe used in the southern blot.

We have added the requested information to the figure legend as shown in highlight in lines 344-346.

  • The reviewer suggested to remove letter A from each bar in figure 3.

Thanks for the suggestion. We have removed the letter A from this figure and added a few words to the figure’s legend to make sure that the legend is in line with the figure as highlighted in lines 370-372.

  • The reviewer suggested to move Figure 7 and 8 to supplementary information.

We have moved Figures 7 and 8 to the supplementary section as requested, along with the legends.

  • The reviewer asked why the sensitivity of mutant noxa-im-5 to H2O2 is quite different from that of the other noxa mutants.

Sorry, at this stage, we can’t be 100% sure what could have caused this. It could be several things, but we would not want to speculate without a solid proof.

  • The reviewer suggested to add a sentence to the figure legends on how to read the letters as letters A, B, C, D and E in figures as they are not easy to follow.

To address this concern, we have added an explanation to the figure legends as highlighted in Lines 308-309 for Figure 1, Lines 370-372 for Figure 3, Lines 381-383 for Figure 4, Lines 391-393 for Figure 5, and Lines 402-404 for Figure 6.

  • The reviewer asked to add the experimental limitation regarding Figure 9 (now Figure 7) in the Figure legend.

We have added the explanation for the experimental limitation to the Figure 7 legend as highlighted in Lines 444-449.

  • The reviewer asked to carefully check English throughout the manuscript.

We have carefully checked English throughout the manuscript.

Reviewer 2 Report

The authors have responded to my suggestions, and I have no further comment for this manuscript.

Author Response

Thank you for the time you spent reviewing this manuscript.

Round 2

Reviewer 1 Report

The manuscript by Zhu et al., describes the importance of NADPH oxidases from Verticillium dahliae in virulence.

The fungus name in the title should be written Verticillium dahliae since it is the species name, first word capitalized and second word with small caps.

I am having difficulties with the statistical analysis representation. I asked authors to give a small explanation in figure legends about the meaning of the letters on top of the columns from graph bars or on top of points from line graphs The sentence added by authors “Letters A and B on top of each figure represent figure numbers while letters on top of bars and lines refer to statistical difference significance between the treatments” does not answer my question and can be removed. What in my opinion is missing is an explanation about what was compared, for instance in graph from figure 1A, and gave differences represented by letter A (first column of the graph) and B and C (second column of the graph).

Line 331, please correct the supplementary figures referred in here according to authors alterations.

Since noxa mutants present no differences in cell wall biosynthesis and in resistance to oxidative and osmotic stresses, maybe these analyses do not justify two independent sections in the manuscript.

In my opinion results section 3.7 and 3.8 should be combined.

Supplementary Figure S2, S5 and S6 still contain information about noxb-complemented strain that is no longer referred in the figure legends nor in the paper.

Author Response

Response to Reviewers’ Comments

Please note that all changes requested by reviewer are now shown in highlight throughout the manuscript.

  • The reviewer asked to correct the species name in the title.

Thanks for your comment. Please note that we have corrected the species name in the title.

  • The reviewer asked to remove previous sentences about statistical part in figure legend and add a sentence to the figure legends on how to read the letters.

To address this concern, we have added a new explanation sentence to the figure legends as shown in highlights. To briefly explain this, we have done statistical analysis followed by an LSD post hoc test to determine the significant differences among variable means as shown in different letters on top of each bars/points in the graphs.   

  • The reviewer asked to correct the supplementary figures referred in Line 331 according to authors alterations.

We have made the correction as shown in highlight in lines 332.

  • The reviewer suggested to combine results from section 3.7 and 3.8.

Thanks for the suggestion. We have combined these two sections.

  • The reviewer pointed out that Supplementary Figures S2, S5 and S6 still contain information about noxb-complemented strain which must be removed.

We have removed information about noxb mutants in the new supplementary files.

This manuscript is a resubmission of an earlier submission. The following is a list of the peer review reports and author responses from that submission.

Round 1

Reviewer 1 Report

The manuscript by Zhu et al., describes the importance of NADPH oxidases from Verticillium dahlia in virulence. The fungus genome contains 3 nox genes whose expression varies between highly and weakly aggressive isolates upon elicitation with extracts from different potato tissues. From the 3 nox genes, noxA and noxB deletion results in reduced virulence with both being essential for infection. However, noxB mutants display alterations in cell wall integrity and are much more resistant to oxidative stress.

Results depicted are well presented and conclusions are supported by the presented results. However, discussion of the paper needs to be improved.

Major points:

NoxA and NoxB mutants have a very similar phenotype in terms of infection and disease severity indicating that both genes are needed for V. dahlia virulence. Looking at the expression profile of noxB in highly versus weakly aggressive strains it would be expected that deletion of this gene would induce a decrease in virulence, as observed in here. In contrast, the phenotype of the noxA mutant was a surprise since this gene is not so much induced upon infection. Moreover, is it possible that NoxB is able to compensate for noxA deletion, as thus noxA mutant has no defects in cell wall neither alterations in sensitivity to H2O2. Please had some discussion about the ability or not of compensation between the different NOX proteins.

How do authors explain that the most resistant strains to H2O2 are the less virulent ones? Isn’t this counterintuitive? It is suggested that ROS produced by NOXB or NOXA are essential for penetration and infection. However, why doesn’t noxa replace for noxb and vice-versa? Is the subcellular location of NOX proteins known?

How do authors explain the differences observed in Figure 5 and 6? Are they related to a more resistant potato cultivar?

Letters A, B, C, D and E in figures are not easy to follow. Since genes are also noxA, B and C it looks to me that another representation for the statistical analyses could be chosen. Please alter this point accordingly if you agree and add a sentence, in figure legend, explaining how to read the letters code in each figure.

Minor points:

Please correct the name of the fungus in title.

Subsections of results are not in the same style as the ones in material and methods.

Line 352, this sentence is already above.

Line 370, is noxc instead of noxa

Line 373, what is present in lane 2? Which probe was used in here? What is the ectopic control NoxA-Ect-3? Is it an overexpression strain of NoxA?

Define WC-water control

It looks important to me to define AUDPC of infection and disease severity

Figure 3, why do authors add A to every bar? What does this means? Is it needed?

Figure 8, since results from figure 8D are referred in the text before results depicted in figures 8A-C maybe osmotic sensitivity could appear as figure 8A.

The sensitivity of mutant noxa-im-5 to H2O2 is quite different from that of the other noxa mutants. Please comment this observation.

Figure 9 and 10, the complemented transformants are not depicted in here.

Why do the plates of 21 days after look so small comparing to the other conditions? Could this be improved?

There are several references using the first author instead as a number throughout the discussion.

Please carefully check English throughout the manuscript.

Reviewer 2 Report

The manuscript "NOX-Family Genes Are Important For Verticillium Dahliae’s Penetration Ability And Virulence" characterized the gene functions of NADPH oxidase (NoxA, NoxB, and NoxC) of Verticillium dahliae on potato. The introduction provided clear overview and the experimental design was standard in general, but there were many unclear presentation of the results, with some essential comparisons:

Major problems

(1) The weakness of lacking complemenation NoxB and NoxC strains 

This study constructed the inserted mutants (instead of target gene deletion mutants) for three genes, and the construction map was unclear. The authors seemed to mention about the insertion sites for each gene mutant (line 327), but a clear construction illustration would be required to help readers understand the mutation background, as well as to provide a knowledge base of Southern blot (Figure 2G-J). Otherwise, the Southern blot makes no sense. Providing only the restriction enzymes in the Table 2 without DNA ladders nor mutant construction illustration, readers would not be able to tell if the mutation was indeed correct or not. Moreover, without a construction illustration, the PCR results may be hard to understand and it appears to readers that Hyg detection in the Figure 2D has different sizes among lanes (And no one knows the size of DNA ladder without labels!).

Other than these uncareful presentations in the Results, the lack of complemented NoxB and NoxC strains in many parts of experiments become a major problem of this study. In Figure 3, it appears that the authors only have one ectopic strain for NoxA and one for NoxC. However, only the results of NoxA were supported by mutants and complemented strains (through Figure 4 to 10). The results of NoxC on the pathogenicity were supported by mutants and complemented strains (Figure 4 to 6, not for Figure 7 to 10). None of the results for NoxB was supported by complemented strains. 

(2) Unclear of biological replicates and experimental repeats, unclear presentation of pathogenicity results

The authors mentioned about n=3 in Figure 1, and the statistics appeared to be good. But if the average was based on 3 data points, it becomes suspicious if the authors ever repeat the experiments? If the authors repeated for three times to confirm the phenomenon was reproducible, shouldn't there be 9 data points? Or the n=3 means the authors repeated three times with only one biological replicate in each time?

In the Figure 4 to 6, the authors showed different mutants in different years. The Figure 4 contains results for NoxA and NoxC in 2016, Figure 5 contains results of NoxA and NoxB in 2017, and Figure 6 contains results of NoxA and NoxB in 2018. Why was the NoxC excluded from experiments in 2017 and 2018? Why the experiments of NoxA and NoxB in 2017 and 2018 separated for analysis? Instead combine to see the overall effects? Without a comprehensive inclusion for all gene mutants and ectopic strains (see comment 1), and without a comprehensive inclusion for all strains in the same situation in the same time/year, the results were not solid. For instance, the current presentation suggested that the authors only conducted "once" for NoxC mutants in 2016. 

(3) Literature review and comparison

The VdNoxB has been studied and published in PLoS Pathogen 2016 as the author cited in [63]. But the authors did not provide comprehensive review and comparison of their results to this previously studied literature. What is the novelty of this study? As the PLoS Pathogen paper indicated that VoxB is required for pathogenicity on cotton, while the authors pointed out that their NoxB was not required for pathogenicity on potato, the difference could be true (or false) only if the authors include the results using target gene deletion mutants and complemented strains. Current presentation based on imperfect experimental supports will raise more questions for VdNoxB.

One minor point would be that although the authors claimed no significant difference for the growth rate of NoxC mutants, but the colony diameter in Figure 3 appeared to be consistently smaller than the wild type. The authors may want to confirm if the description or phenotype is correct or not.

In summary, this study aimed to elucidate the function of VdNoxA/B/C on potato, but the scientific soundness for VdNoxB/C was insufficient. The authors may want to conduct additional research to complete the tasks, or if the authors would like to focus on the VdNoxA and rephrase the manuscript just for VdNoxA. Additional works would be needed.